| Open Peer Review | Mycology | Methods and Protocols

# The Fourier-transform infrared spectroscopy-based method as a new typing tool for *Candida parapsilosis* clinical isolates

Elena De Carolis,[1] Brunella Posteraro,[2,3] Benedetta Falasca,[2] Bram Spruijtenburg,[4] Jacques F. Meis,[4,5] Maurizio Sanguinetti[1,2]

**ABSTRACT** The Fourier-transform infrared spectroscopy-based IR Biotyper is a straightforward typing tool for bacterial species, but its use with *Candida* species is limited. We applied IR Biotyper to *Candida parapsilosis*, a common cause of nosocomial bloodstream infection (BSI), which is aggravated by the intra-hospital spread of fluconazole-resistant isolates. Of 59 *C. parapsilosis* isolates studied, $n = 56$ (48 fluconazole-resistant and 8 fluconazole-susceptible) and $n = 3$ (2 fluconazole-resistant and 1 fluconazole-susceptible) isolates, respectively, had been recovered from BSI episodes in 2 spatially distant Italian hospitals. The latter isolates served as an outgroup. Of fluconazole-resistant isolates, $n = 40$ (including one outgroup) harbored the Y132F mutation alone and $n = 10$ (including one outgroup) harbored both Y132F and R398I mutations in the *ERG11*-encoded azole-target enzyme. Using a microsatellite typing method, which relies on the amplification of genomic short tandem repeats (STR), two major clusters were obtained based on the mutation(s) (Y132F or Y132F/R398I) present in the isolates. Regarding IR Biotyper, each isolate was analyzed in quintuplicate using an automatic (i.e., proposed by the manufacturer's software) or tentative (i.e., proposed by us) cutoff value. In the first case, four clusters were identified, with clusters I and II formed by Y132F or Y132F/R398I isolates, respectively. In the second case, six subclusters (derived by the split of clusters I and II) were identified. This allowed to separate the outgroup isolates from other isolates and to increase the IR Biotyper typeability. The agreement of IR Biotyper with STR ranged from 47% to 74%, depending on type of cutoff value used in the analysis.

**IMPORTANCE** Establishing relatedness between clinical isolates of *Candida parapsilosis* is important for implementing rapid measures to control and prevent nosocomial transmission of this *Candida* species. We evaluated the FTIR-based IR Biotyper, a new typing method in the *Candida* field, using a collection of fluconazole-resistant *C. parapsilosis* isolates supposed to be genetically related due to the presence of the Y132F mutation. We showed that IR Biotyper was discriminatory but not as much as the STR method, which is still considered the method of choice. Further studies on larger series of *C. parapsilosis* isolates or closely related *Candida* species will be necessary to confirm and/or extend the results from this study.

**KEYWORDS** FTIR spectroscopy, IR Biotyper, *C. parapsilosis*, fluconazole resistance, Erg11p mutation, microsatellite genotyping, STR

Together with some other *Candida* species (including *C. albicans*), *Candida parapsilosis sensu stricto*, which belongs to the *C. parapsilosis* species complex (also including *C. orthopsilosis* and *C. metapsilosis*), remains a leading cause of nosocomial bloodstream infection (BSI) worldwide (1). This species (hereafter referred to as *C. parapsilosis*) continues to be dominant and/or associated with neonatal or pediatric hospital settings (2), particularly in Southern Europe (i.e., Mediterranean regions), Asia, and South America

Address correspondence to Maurizio Sanguinetti, Maurizio.sanguinetti@unicatt.it.

Elena De Carolis and Brunella Posteraro contributed equally to this article. Author order was determined alphabetically.

The authors declare no conflict of interest.

See the funding table on p. 9.

(3–11). Due to the propensity to form biofilm on medically implanted devices (including intravascular catheters) and the persistence in hospital environments (12), *C. parapsilosis* is increasingly involved in nosocomial outbreaks (13–19). This has stimulated the (re)search for typing methods to support epidemiological investigations, which are crucial to control and prevent the spread of *C. parapsilosis* (and other *Candida* species) within hospitals.

Since its development in 2006 (20), microsatellite genotyping, which is based on the amplification of genomic short tandem DNA repeats (STRs), has become the method of choice for assessing relatedness between *C. parapsilosis* clinical isolates (21). Using STR (hereafter used instead of microsatellite), studies (4, 6, 8, 16–18) have shown that outbreak or persistent clonal isolates of *C. parapsilosis* clustered according to being resistant or susceptible to fluconazole and that fluconazole-resistant isolates harbored the same mutation (i.e., Y132F) in the *ERG11*-encoded azole-target fungal enzyme (2). No link between the presence of Erg11p mutation and STR-based clustering of *C. parapsilosis* isolates was shown elsewhere (11, 22), suggesting that STR may not be discriminatory enough.

The Fourier-transform infrared (FTIR) spectroscopy is a rapid, high-throughput, and simple phenotypic method that captures differential vibrational modes of IR light-absorbing chemical bonds directly on the cell structure of different microbial (bacterial) species (23). As a reflection of chemical cell composition, a specifically generated IR spectrum represents the IR absorption signature (or fingerprint) for a given microbe. The carbohydrates' IR region (1,300–800 cm$^{-1}$) covers most of the spectral differences between isolates of the same microbial species. While FTIR (i.e., FTIR-based IR Biotyper; Bruker Daltonics, Bremen, Germany) is currently adopted for bacterial typing (24), the application of IR Biotyper on *Candida* species is currently limited (25, 26). Therefore, we aimed to evaluate IR Biotyper as a new typing method for *C. parapsilosis* clinical isolates and compare results with those obtained with the STR method. Excluding 3 outgroup isolates, *n* = 56 (48 fluconazole-resistant and 8 fluconazole-susceptible) of the 59 *C. parapsilosis* isolates studied here were BSI isolates from a previously reported (4) and still ongoing large Italian-hospital survey.

This work was presented, in part, at the 32nd European Congress of Clinical Microbiology and Infectious Diseases (ECCMID) held in Lisbon, Portugal (23–26 April 2022).

## RESULTS

We used 56 *C. parapsilosis* isolates that had been recovered from hospital-acquired BSI episodes over a 2014-to-2022-year period in Rome (Lazio region), Italy. The fluconazole-resistant isolates harbored point mutation(s) in the *ERG11* gene, which translated to the Erg11p amino acid substitution Y132F (48 isolates) or R398I (9 out of 48 isolates). As an outgroup, we used three (two fluconazole-resistant and one fluconazole-susceptible) *C. parapsilosis* isolates that had been recovered, in 2022, from hospital-acquired BSI episodes in Arezzo (Tuscany region), Italy. Of fluconazole-resistant isolates, one had the Y132F mutation, and one had both Y132F and R398I mutations. None of fluconazole-susceptible isolates had an azole resistance-associated Erg11p mutation. Details about the study isolates are provided in Table S1.

STR typing based on a panel of 6 (3 trinucleotide and 3 hexanucleotide) markers allowed to group 59 *C. parapsilosis* isolates in 2 major clusters (Fig. 1). Thirty-four Y132F-harboring isolates in cluster I and eight Y132F/R398I-harboring isolates in cluster II were clonal, i.e., had the same genotype. The remaining Y132F-harboring isolates (six, including one outgroup) were related to cluster I, whereas the remaining Y132F/R398I-harboring isolates (two, including one outgroup) were related to cluster II. Except for two (including one outgroup), seven out of nine fluconazole-susceptible isolates had a clearly different genotype than the isolates from clusters I and II.

Results from the IR Biotyper analysis for 59 *C. parapsilosis* isolates are presented in Fig. 2 and 3, whereas representative spectra of isolates classified as resistant or susceptible to fluconazole are shown in Fig. 4. Using an optimized cutoff value (0.967), which was

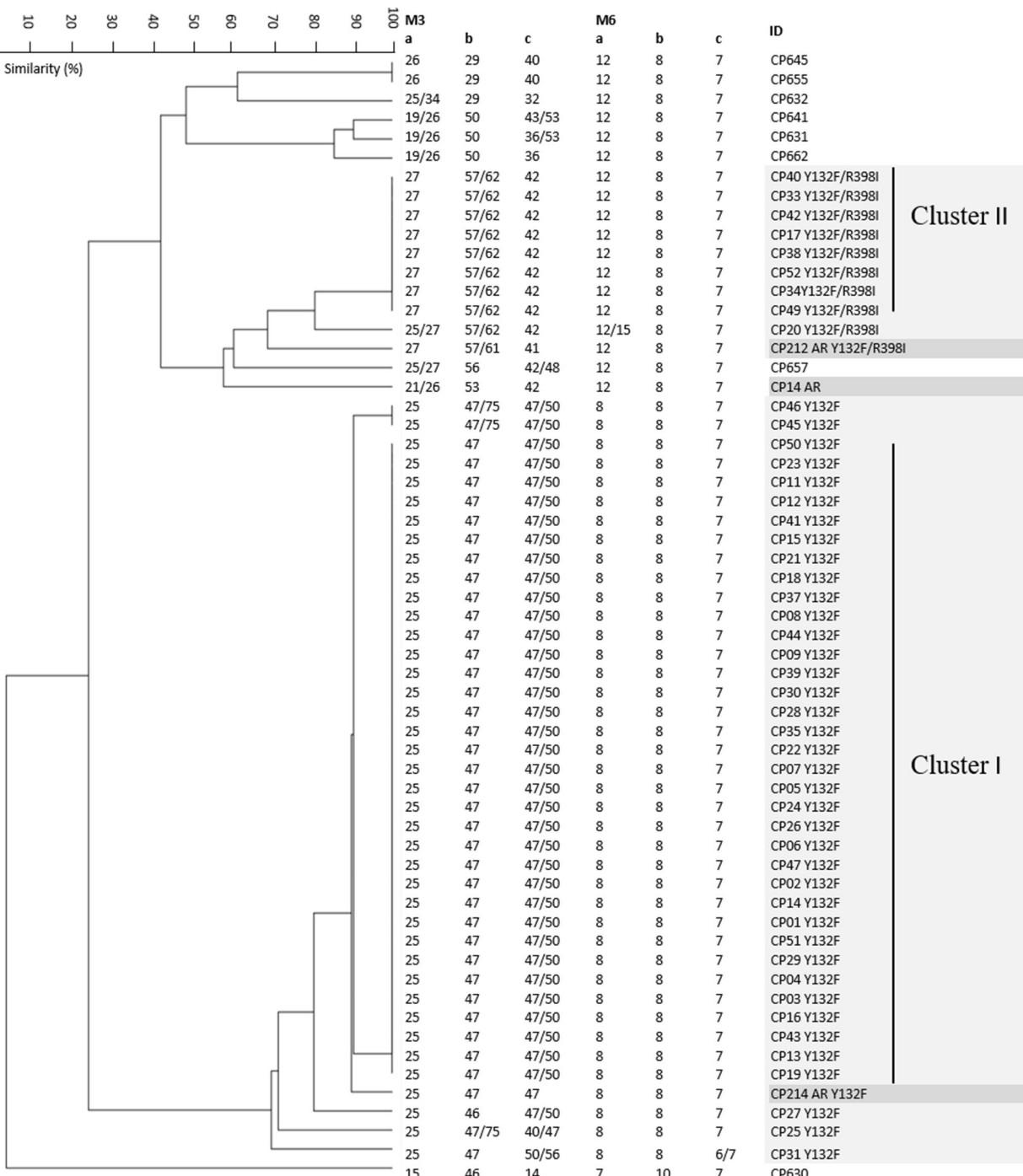

**FIG 1** STR typing of *C. parapsilosis* isolates with (*n* = 50) or without (*n* = 9) azole-resistance associated Erg11p mutations. Dendrogram was based on the analysis of STR markers, three trinucleotide (M3) and three hexanucleotide (M6), which generated two series of microsatellite genotypes. The scale bar (visible in the upper left corner) indicates the similarity percentages. Two major clusters, namely, STR I and STR II, were identified.

automatically generated by the software, isolates were grouped into four (two major and two minor) clusters (Fig. 2). The major clusters comprised Y132F-harboring isolates (40, including 1 outgroup; cluster I) and Y132F/R398I-harboring isolates (10, including 1 outgroup; cluster II), respectively. The nine fluconazole-susceptible isolates were distributed across clusters [cluster I, one isolate; cluster II, three isolates; cluster III, three isolates; and cluster IV, two isolates (including one outgroup)]. Using a tentative cutoff value (0.995), which was based on our experimental input, isolates were grouped in six

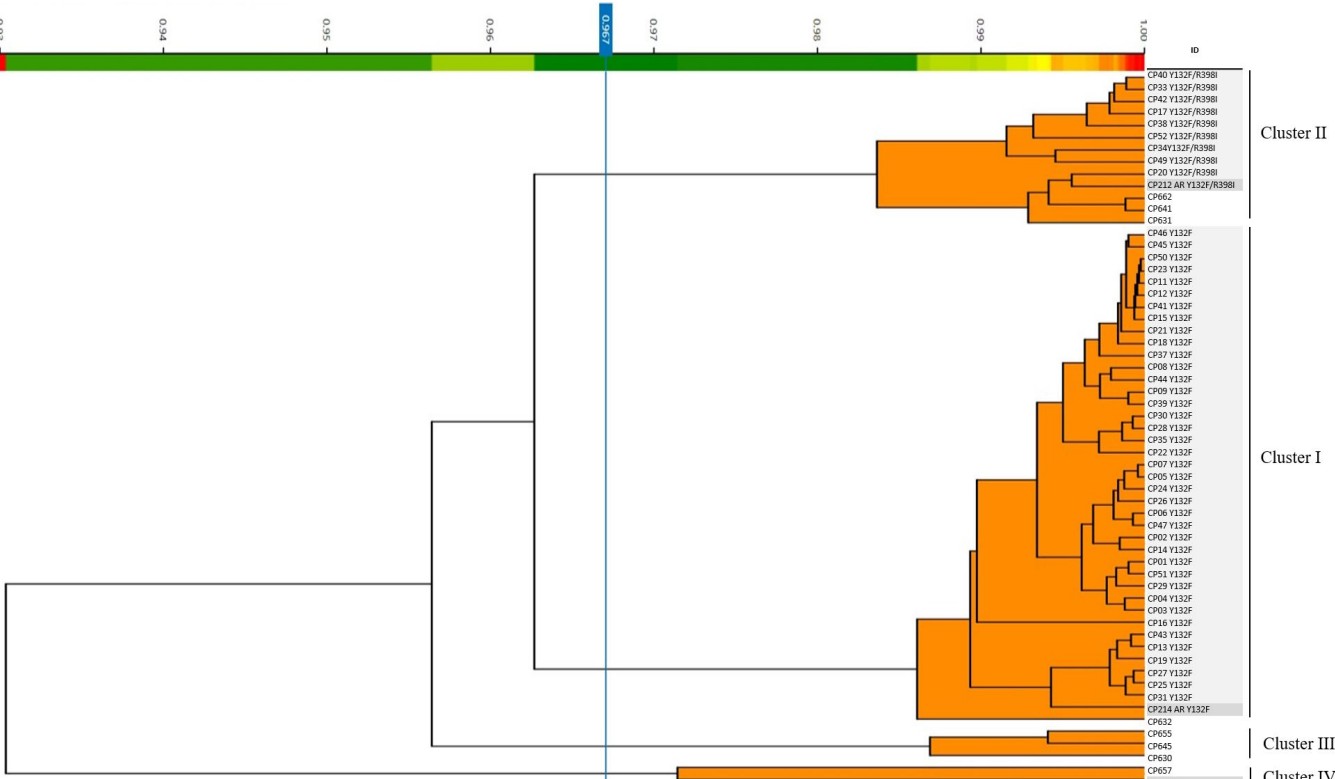

**FIG 2** FTIR typing of *C. parapsilosis* isolates with (*n* = 50) or without (*n* = 9) azole-resistance associated Erg11p mutations. Dendrogram was based on the IR Biotyper analysis of spectra obtained from replicate measurements for each isolate. The vertical line represents the automatic cutoff value (0.967). Four clusters (two major and two minor), namely, IR I to IR IV, were identified.

(sub)clusters, of which four derived from the split of cluster I (referred to as Ia to Id) and two derived from the split of cluster II (referred to as IIa and IIb) (Fig. 3). This allowed to separate the three (one Y132F/R398I-harboring, one Y132F-harboring, and one

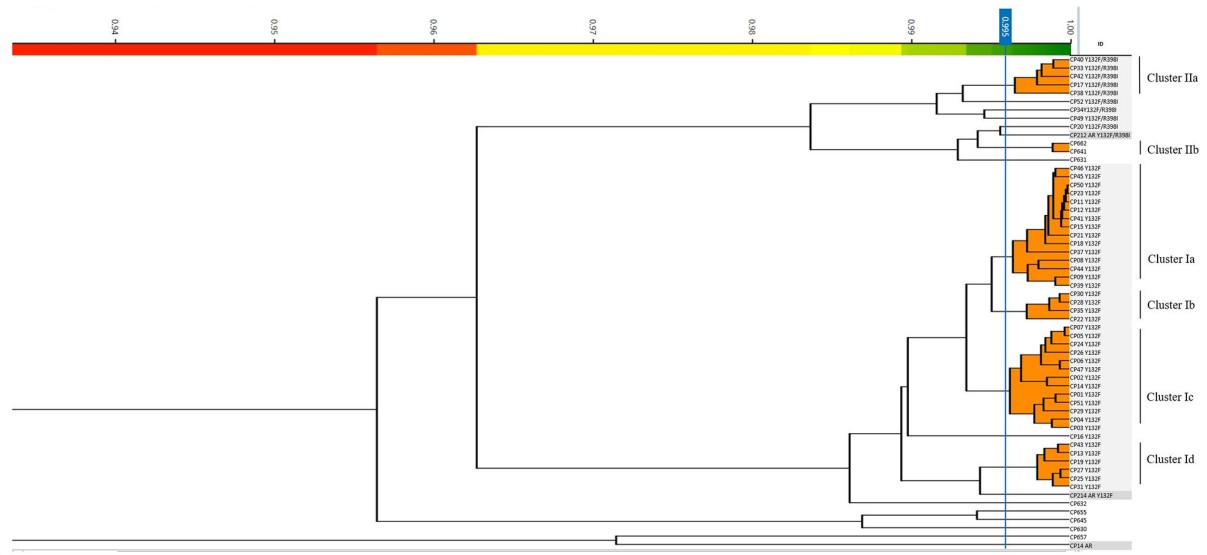

**FIG 3** FTIR typing of *C. parapsilosis* isolates with (*n* = 50) or without (*n* = 9) azole-resistance-associated Erg11p mutations. Dendrogram was based on the IR Biotyper analysis of spectra obtained from replicate measurements for each isolate. The vertical line represents the tentative cutoff value (0.995). Six subclusters (derived, respectively, from clusters IR I and IR II; see Fig. 2), namely, IR Ia to IR Id and IR IIa to IR IIb, were identified.

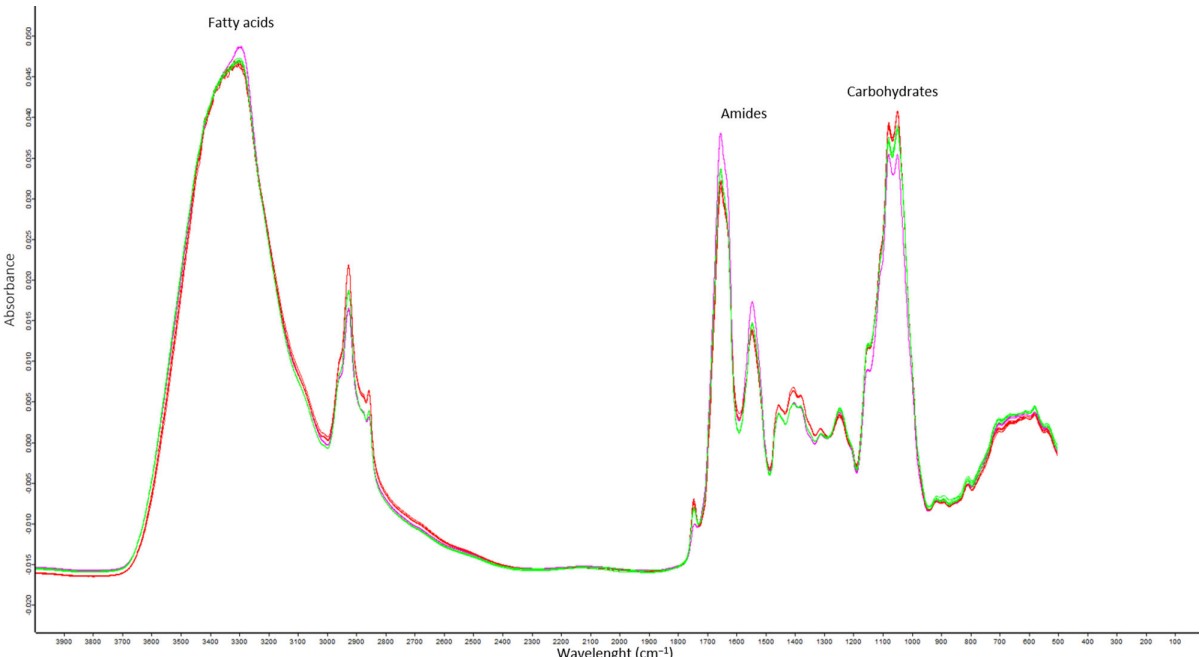

**FIG 4** FTIR spectral profiles of fluconazole-resistant (Y132F or Y132F/R398I mutation(s) harboring) or fluconazole-susceptible *C. parapsilosis* isolates. Different colors (green, pink, or red) indicate the three isolates, respectively. Profiles overlap and only subtle differences in absorbance are visible. Spectra were acquired in the mid-IR (4,000–400 cm$^{-1}$), which includes spectral windows corresponding to the absorption of lipids (3,000–2,800 cm$^{-1}$, dominated by fatty acids), proteins and peptides (1,800–1,500 cm$^{-1}$, dominated by amides), or polysaccharides (1,200–900 cm$^{-1}$, dominated by carbohydrates).

fluconazole-susceptible) outgroup isolates, respectively, from those of clusters Ia, Ib, Ic, and Id (each including Y132F-harboring isolates); cluster IIa (including most Y132F/R398I-harboring isolates); or cluster IIb (including some fluconazole-susceptible isolates).

We compared the two typing methods regarding discriminatory ability, which was measured by the Simpson's index of diversity. The index value was 0.655 [95% confidence interval (CI), 0.520–0.789] for the STR method. When the analysis was performed using the automatic cutoff value, IR Biotyper showed an index value of 0.473 (95% CI, 0.345–0.600), and the agreement with STR was 0.473 (95% CI, 0.209–0.737). When the analysis was performed using the tentative cutoff value, IR Biotyper showed an index value of 0.839 (95% CI, 0.775–0.904), and the agreement with STR was 0.739 (95% CI, 0.524–0.954).

## DISCUSSION

We assessed the comparability of IR Biotyper and STR for typing *C. parapsilosis* using a large collection of clinical isolates, with approximately 85% of them having a fluconazole-resistant phenotype explained by the well-known Erg11p Y132F mutation. Both methods were able to identify clusters of isolates according to the presence of the Y132F mutation (alone or in combination with the R398I mutation). These clusters were separated from those formed by fluconazole-susceptible isolates. The agreement of IR Biotyper with the STR method ranged from 47% to 74%, depending on whether an automatic (i.e., proposed by the manufacturer's software) or tentative (i.e., proposed by us) cutoff value, was used for the analysis.

To our knowledge, only two studies to date have investigated IR Biotyper as a typing tool for *Candida* species (25, 26). In the first study (25), using 28 *Candida auris* isolates from patients at a hospital in Los Angeles, CA, the IR Biotyper allowed the isolates to form 2 clusters, 1 comprising 27 isolates from the South African clade and 1 comprising 1 isolate from the South Asian clade. In that study (25), IR Biotyper results confirmed those obtained with the whole-genome sequencing (WGS) method, which was not

actually used as a comparator because of different output measures between IR Biotyper and WGS methods. The second (larger) study (26), using 96 *C. auris* isolates from 14 geographical areas, found that IR Biotyper was concordant with STR by 33% similarity. This value was slightly lower than that observed (45% similarity) comparing STR with the fungal (*Candida*)-specific ribosomal DNA internal transcribed spacer region sequencing method (26). Compared to the STR method, which grouped the isolates into four clusters corresponding to four (geographically distinct) *C. auris* clades (i.e., East Asian, South Asian, South African, and South American), IR Biotyper separated or grouped isolates regardless of their geographical origin. However, clustering was identical for 80% of South American isolates with all typing methods (also including amplified fragment length polymorphism and MALDI-TOF mass spectrometry) used in that study (26), suggesting that capitalizing the IR Biotyper method necessarily requires species-specific setup efforts.

Despite being in a different microbial context, a study showed that IR Biotyper was equivalent to WGS when applied to extended-spectrum-beta-lactamase-producing *Klebsiella pneumoniae* isolates. The study (27) analyzed 18 isolates from 14 patients and found concordance between IR Biotyper and WGS for all but 1 isolate using the cutoff value generated by the manufacturer' software after 4 technical replicates (0.037); concordance between the methods reached 100% using the cutoff value generated by the manufacturer' software after 12 technical replicates (0.046). While it is intuitive that changes in the cutoff value affect the discriminatory power of the IR Biotyper, the manufacturer leaves users the freedom to adjust the analysis acting on the cutoff value to be used to define clusters. In the case of *K. pneumoniae*, the IR Biotyper manufacturer has made available a list of recommended cutoff values based on user input (27).

Consistent with these observations, moving the cutoff value from 0.967 to 0.995 increased the IR Biotyper discriminatory ability with our *C. parapsilosis* isolates. Accordingly, the originally identified clusters I and II were split into subclusters, which allowed the two outgroup isolates, one with the Y132F mutation and the other with Y132F/R398I mutations, to be separated, respectively, from the groups of isolates that harbored the same mutation(s) but originated from BSI episodes in a different, spatially distant Italian region (Lazio instead of Tuscany) hospital. Otherwise, further discrimination within the original clusters I and II underscored greater typeability of IR Biotyper than the STR method, which could accentuate the phenotype-level differences within a class of (Y132F-harboring) fluconazole-resistant isolates with reduced genetic diversity (28). Owing to the Y132F mutation, these isolates had a higher propensity to spread in a clonal mode than fluconazole-susceptible isolates, which would account for their human/environmental persistence as well as their association with candidemia or other forms of invasive candidiasis (28). It is noteworthy that Y132F-harboring isolates were always discriminated from those with the Y132F detected in combination with the R398I mutation, which does not confer azole resistance *per se* but may offset the fitness cost of acquiring the Y132F mutation (29).

Our findings apparently contrast with those obtained by Vatanshenassan et al. (26), who applying IR Biotyper to *C. auris* isolates showed that neither IR Biotyper nor the other four typing methods used had greater than 50% agreement with STR. In our study, we exceeded this threshold although we were still quite far from 100%. There are some (dis)similarities between *C. auris* and *C. parapsilosis* that could help explain the differences between the two studies. First, mutations known to affect azole resistance in *C. parapsilosis* (and other *Candida* species) are also present in fluconazole-resistant isolates of *C. auris* (30), but these mutations vary between the geographical clades to which *C. auris* isolates belong (i.e., F126L in South Africa, Y132F in Venezuela, and Y132F or K143R in India/Pakistan). Unlike the Vatanshenassan et al.'s study (26), we studied a collection of fluconazole-resistant *C. parapsilosis* isolates supposed to be genetically related due to the presence of the Y132F mutation. Accordingly, harboring the Y132F mutation would imply a cell surface specifically enriched with glycoproteins (e.g., those encoded by the agglutinin-like sequence gene family), which are consistent with the

adhesive capacities or morphological properties (2), biofilm formation (31), or increased fitness (28) shown by *C. parapsilosis* isolates. These attributes are significantly reduced in *C. auris* (32). The IR Biotyper results matched better the STR results in our study than in the Vatanshenassan et al.'s study (26), but it will be necessary to confirm our results on larger series of *C. parapsilosis* isolates or closely related *Candida* species (2).

We did not compare IR Biotyper to WGS. The reason is that the STR method used in this study had been optimized to improve its discriminatory power (33) and subsequently adopted in studies dealing with fluconazole-resistant *C. parapsilosis* (4, 8, 28). However, it is plausible that using WGS to validate the comparison of IR Biotyper with the STR method in this study would have strengthened our results. Due to the retrospective nature of our study, we could not use IR Biotyper to show an epidemiological link between the *C. parapsilosis* isolates studied, nor could we know that our isolates had been part of an invasive candidiasis outbreak. However, from a methodological point of view, this study represents the third time IR Biotyper was applied to the *Candida* field. Because phenotyping may be influenced by differences between *Candida* species, major efforts are necessary before an IR Biotyper-based method can be applied to all medically relevant *Candida* species.

In conclusion, we showed that the IR Biotyper has the potential to become a reliable typing tool for *C. parapsilosis*. Its rapidity and simplicity make it attractive particularly if compared to WGS, which has become prominent in today's clinical microbiology laboratory. Although further studies are needed to confirm our findings, we believe this study may prompt future efforts to improve the applicability of typing methods such as the IR Biotyper for *C. parapsilosis* and other pathogenic *Candida* species.

## MATERIALS AND METHODS

### Yeast isolates

We retrospectively studied 59 *C. parapsilosis* isolates, of which 56 were obtained in the years 2014 to 2022, from BSI patients at the Policlinico Universitario A. Gemelli IRCCS of Rome, Italy, and 3 were obtained in the year 2022, from BSI patients at the San Luca Hospital of Arezzo, Italy (see Table S1). The last three isolates were included as an outgroup control. All isolates had previously been characterized for the presence (fluconazole-resistant isolates, $n = 50$) or absence (fluconazole-susceptible isolates, $n = 9$) of azole resistance-associated point mutations in the *ERG11* gene (2). The isolates were stored at −80°C and subcultured before testing.

### STR analysis

Isolates were grown at 37°C on Sabouraud dextrose agar (SDA) plates (Kima, Padua, Italy) for 48 h prior to DNA extraction. As previously described (4), DNA (~1 ng) from each isolate was used for multiplex PCRs, which were performed with specific sets of primers for six (three trinucleotide and three hexanucleotide) STR markers (33). The PCR amplicons were diluted 1:200 in water, and 10 µL dilution samples was analyzed on a 3500xL Genetic Analyzer (Applied Biosystems, Foster City, CA). The number of copies for each STR marker was determined using the GeneMapper software 5 (Applied Biosystems), whereas the allele numbers' combination for the six loci allowed to determine each isolate's genotype. According to previous definitions (4), isolates with the same alleles for all six loci had the same genotype (clonal isolates), whereas isolates with genotypes that differed in only one of the six loci were closely related. The relatedness of isolates was calculated with the unweighted pair group method with arithmetic mean (UPGMA) algorithm using the Applied Maths BioNumerics software v.7.6.1 (https://www.applied-maths.com/) and visualized in a dendrogram.

## FTIR analysis

Isolates were grown at 37°C on SDA plates for 48 h prior to sample preparation. According to the IR Biotyper manufacturer's instructions, a 10 µL inoculation loopful of yeast cells was suspended in 50 µL of 70% ethanol solution. The suspension was homogenized by vortexing with metal beads (Bruker Daltonics) in 1.5-mL microcentrifuge tubes. An additional volume (50 µL) of deionized water was added to reach a suspension's final volume of 100 µL. After vortexing, 15 µL of the suspension was spotted in quintuplicate (technical replicates) on a silicon sample plate (Bruker Daltonics) and dried at 37°C before analysis. Three independent experiments (biological replicates) were performed to encompass the inter-run variation, and quality controls, which consisted of two Bruker infrared test standards, were included in duplicate in each run.

Spectra were analyzed (in the 1,300–800 $cm^{-1}$ carbohydrate region) using the IR Biotyper system (Bruker Daltonics) with default analysis settings, as previously described [26]. Spectra were checked for quality control criteria, such as absorbance intensity, signal-to-noise ratio, and water vapor disturbance. Spectra that did not meet these criteria were excluded from the analysis. For each isolate, spectra were preprocessed using second derivation followed by vector normalization [34]; then, an average spectrum was obtained with 5 spectra of each replicate (15 FTIR measurements in total) and was used as a representative spectrum for cluster analysis. Using IR Biotyper v3.1.2 software, which automatically generates a cutoff value to define the distance to which the spectra are in the same cluster, the similarity between spectra was assessed with the average linkage hierarchical clustering algorithm, which calculates the Pearson correlation coefficient of spectra. The algorithm allowed to construct a dendrogram, which gives a tree-like overview of the spectral relationships. In addition to software-generated cutoff value, we used a self-generated (called tentative) cutoff value, defined as the lowest cutoff value at which separation of presumably unrelated (i.e., outgroup) from related isolates was possible.

The above region of 1,300–800 $cm^{-1}$ includes the IR spectral window of polysaccharide absorption (1,200–900 $cm^{-1}$, dominated by microbial cell wall carbohydrates) and, in part, the IR spectral window corresponding to a fingerprint region (900–700 $cm^{-1}$, based on specific patterns but not attributed to microbial cell components) [23, 34]. Additionally, spectra were analyzed in other IR spectral windows within the mid-IR (4,000–400 $cm^{-1}$), corresponding to the absorption of lipids (3,000–2,800 $cm^{-1}$, dominated by fatty acids), proteins and peptides (1,800–1,500 $cm^{-1}$, dominated by amides), or a mixture of biomolecules (1,500–1,200 $cm^{-1}$) [23]. According to previous findings [34], we confirmed the 1,300–800 $cm^{-1}$ spectral region as the main contributor to the discriminatory potential of IR Biotyper (data not shown).

## Discriminatory power and concordance

For each typing method (i.e., STR or IR Biotyper), the discriminatory ability was calculated using the Simpson's index of diversity, which indicates the probability that two unrelated isolates randomly chosen will be assigned to different clusters by the method. The higher the value of the index (>0 to ≤1), the more discriminant is the method. The concordance of STR and IR Biotyper was calculated using the adjusted Wallace coefficient (http://www.comparingpartitions.info/?link=Tut4), which indicates how much one of the two methods agrees with the other. Values of 1 and 0 indicate perfect or no agreement between methods, respectively.

## ACKNOWLEDGMENTS

We thank Eva Maria Parisio and Giulio Camarlinghi (San Luca Hospital of Arezzo, Italy) for providing *C. parapsilosis* isolates.

This research was supported by EU funding within the NextGenerationEU-MUR PNRR Extended Partnership initiative on Emerging Infectious Diseases (Project no. PE00000007, INF-ACT) and by the Canisius-Wilhelmina Hospital (Grant CWZ_001421).

## AUTHOR AFFILIATIONS

[1]Dipartimento di Scienze di Laboratorio e Infettivologiche, Fondazione Policlinico Universitario A. Gemelli IRCCS, Roma, Italy

[2]Dipartimento di Scienze Biotecnologiche di Base, Cliniche Intensivologiche e Perioperatorie, Università Cattolica del Sacro Cuore, Roma, Italy

[3]Dipartimento di Scienze Mediche e Chirurgiche Addominali ed Endocrino Metaboliche, Fondazione Policlinico Universitario A. Gemelli IRCCS, Roma, Italy

[4]Radboudumc/Canisius Wilhelmina Hospital Center of Expertise for Mycology, Nijmegen, the Netherlands

[5]Department I of Internal Medicine and ECMM Excellence Center for Medical Mycology, Faculty of Medicine, University Hospital Cologne, Cologne, Germany

## AUTHOR ORCIDs

Maurizio Sanguinetti (iD) http://orcid.org/0000-0002-9780-7059

## FUNDING

| Funder | Grant(s) | Author(s) |
| --- | --- | --- |
| European Union | PE00000007 | Maurizio Sanguinetti |
| Canisius-Wilhelmina Hospital | CWZ_001421 | Jacques F. Meis |

## AUTHOR CONTRIBUTIONS

Elena De Carolis, Conceptualization, Methodology, Supervision, Validation, Writing – review and editing | Brunella Posteraro, Data curation, Formal analysis, Supervision, Validation, Writing – original draft | Benedetta Falasca, Methodology, Investigation | Bram Spruijtenburg, Validation, Writing – review and editing | Jacques F. Meis, Validation, Writing – review and editing | Maurizio Sanguinetti, Formal analysis, Funding acquisition, Validation, Writing – review and editing

## ADDITIONAL FILES

The following material is available online.

### Supplemental Material

**Table S1 (Spectrum02388-23-s0001.pdf).** Characteristics of *Candida parapsilosis* (*sensu stricto*) isolates included in the study

### Open Peer Review

**PEER REVIEW HISTORY (review-history.pdf).** An accounting of the reviewer comments and feedback.

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
