## [Reviewer comments · Microbiology Spectrum]

Microbiology Spectrum

The Fourier-Transform Infrared Spectroscopy Based Method as a New Typing Tool for *Candida parapsilosis* Clinical Isolates

Elena De Carolis, Brunella Posteraro, Benedetta Falasca, Bram Spruijtenburg, Jacques Meis, and Maurizio Sanguinetti

Corresponding Author(s): Maurizio Sanguinetti, Fondazione Policlinico Universitario Agostino Gemelli IRCCS

Review Timeline:

Submission Date:	June 7, 2023
Editorial Decision:	July 3, 2023
Revision Received:	July 17, 2023
Accepted:	July 21, 2023

Editor: Renato Kovacs

Reviewer(s): Disclosure of reviewer identity is with reference to reviewer comments included in decision letter(s). The following individuals involved in review of your submission have agreed to reveal their identity: Alessandra Koehler (Reviewer #2)

Transaction Report:

DOI: <https://doi.org/10.1128/spectrum.02388-23>

July 3, 2023

Prof. Maurizio Sanguinetti
Fondazione Policlinico Universitario Agostino Gemelli IRCCS
Microbiology
L.go A. Gemelli 8
Rome, RM 168
Italy

Re: Spectrum02388-23 (The Fourier-Transform Infrared Spectroscopy Based Method as a New Typing Tool for *Candida parapsilosis* Clinical Isolates)

Dear Prof. Maurizio Sanguinetti:

Link Not Available

Sincerely,

Renato Kovacs

Journals Department
Reviewer comments:

Reviewer #1 (Comments for the Author):

„We showed that IR Biotyper was comparable to the STR method, which is still considered the method of choice" - seems to be an inadequate and very strong statement since the results showed 47 to 74% agreement. In my opinion, a comparison with WGS would be justified here. Otherwise, the results are not very convincing.
Results: The authors should provide characteristic spectra for *C. parapsilosis* and discuss the spectral changes between fluconazole-susceptible and fluconazole-resistant strains.

Methods:

How were the replicates treated? Were the spectra averaged, or do the results show individual results?

Were there any differences among the replicates?

Was the ability to form biofilms on solid artificial surfaces tested? It was shown that biofilm formation changes the FTIR spectra, and it is well-known that some *C. parapsilosis* strains have this ability.

It is not clear how the tentative value was determined. I believe it is crucial since it improved the results so significantly.

Reviewer #2 (Comments for the Author):

The article "The Fourier-Transform Infrared Spectroscopy Based Method as a New Typing Tool for *Candida parapsilosis* Clinical Isolates" is relevant, considering the importance of developing techniques for typing microorganisms that are faster and more appropriate in the hospital context. Methods based on spectroscopy are particularly relevant, considering the advantages and numerous possibilities of these techniques. The authors showed that the IR Biotyper is capable of detecting differences in isolates considered clonal by the STR, increasing its discriminatory power in relation to the standard method.

The work is written concisely and clearly and is interesting and relevant. However, I have some aspects to point out that needs to be better explored in the writing, mainly in the Materials and Methods section:

1) Line 139: "To our knowledge, only one study to date has investigated IR Biotyper as a typing tool for *Candida* species (25)" Please include the following article in the Discussion: Contreras DA, Morgan MA. Surveillance diagnostic algorithm using real-time PCR assay and strain typing method development to assist with the control of *C. auris* amid COVID-19 pandemic. *Front Cell Infect Microbiol.* 2022;12:887754. doi: 10.3389/fcimb.2022.887754.

In this article, the authors used the IR Biotyper to genotype 28 hospital isolates of *Candida auris*, using whole genome sequencing as a reference method. In addition, they were also able to use the IR Biotyper to successfully differentiate several yeast species, such as *C. albicans*, *C. glabrata*, *C. auris*, and *Cryptococcus neoformans*.

2) Line 238: "Spectra were acquired (in the 1300-800 cm⁻¹ carbohydrate region) using the IR Biotyper system..."

I know that this is the default of the instrument and also the "fingerprint" region. However, considering that the maximum Simpson's index of diversity value obtained was 0.839, was it considered that analyzing other regions of the spectrum could improve this index?

3) Was any kind of pre-processing used in the spectra (normalization, derivatives...)? Could these pre-processes improve the discriminatory power of the IR Biotyper? I suggest better detailing the chemometric analyzes in the methodology.

4) Line 116: "Using a tentative cutoff value (0.995), which was based on our experimental input..."

The tentative cutoff was suggested by IR Biotyper software? Explain better how the tentative cutoff value was defined. It would be interesting to add this information in the methodology.

Staff Comments:

Preparing Revision Guidelines

Please return the manuscript within 60 days; if you cannot complete the modification within this time period, please contact me. If you do not wish to modify the manuscript and prefer to submit it to another journal, please notify me of your decision immediately so that the manuscript may be formally withdrawn from consideration by Microbiology Spectrum.

„We showed that IR Biotyper was comparable to the STR method, which is still considered the method of choice“ – seems to be an inadequate and very strong statement since the results showed 47 to 74% agreement. In my opinion, a comparison with WGS would be justified here. Otherwise, the results are not very convincing.

Results: The authors should provide characteristic spectra for *C. parapsilosis* and discuss the spectral changes between fluconazole-susceptible and fluconazole-resistant strains.

Methods:

How were the replicates treated? Were the spectra averaged, or do the results show individual results?

Were there any differences among the replicates?

Was the ability to form biofilms on solid artificial surfaces tested? It was shown that biofilm formation changes the FTIR spectra, and it is well-known that some *C. parapsilosis* strains have this ability.

It is not clear how the tentative value was determined. I believe it is crucial since it improved the results so significantly.

The article “**The Fourier-Transform Infrared Spectroscopy Based Method as a New Typing Tool for *Candida parapsilosis* Clinical Isolates**” is relevant, considering the importance of developing techniques for typing microorganisms that are faster and more appropriate in the hospital context. Methods based on spectroscopy are particularly relevant, considering the advantages and numerous possibilities of these techniques. The authors showed that the IR Biotyper is capable of detecting differences in isolates considered clonal by the STR, increasing its discriminatory power in relation to the standard method.

The work is written concisely and clearly and is interesting and relevant. However, I have some aspects to point out that needs to be better explored in the writing, mainly in the Materials and Methods section:

1) Line 139: “To our knowledge, only one study to date has investigated IR Biotyper as a typing tool for *Candida* species (25)”

Please include the following article in the Discussion: **Contreras DA, Morgan MA. Surveillance diagnostic algorithm using real-time PCR assay and strain typing method development to assist with the control of *C. auris* amid COVID-19 pandemic. Front Cell Infect Microbiol. 2022;12:887754. doi: 10.3389/fcimb.2022.887754.**

In this article, the authors used the IR Biotyper to genotype 28 hospital isolates of *Candida auris*, using whole genome sequencing as a reference method. In addition, they were also able to use the IR Biotyper to successfully differentiate several yeast species, such as *C. albicans*, *C. glabrata*, *C. auris*, and *Cryptococcus neoformans*.

2) Line 238: “Spectra were acquired (in the 1300–800 cm⁻¹ carbohydrate region) using the IR Biotyper system...”

I know that this is the default of the instrument and also the “fingerprint” region. However, considering that the maximum Simpson's index of diversity value obtained was 0.839, was it considered that analyzing other regions of the spectrum could improve this index?

3) Was any kind of pre-processing used in the spectra (normalization, derivatives...)? Could these pre-processes improve the discriminatory power of the IR Biotyper? I suggest better detailing the chemometric analyzes in the methodology.

4) Line 116: “Using a tentative cutoff value (0.995), which was based on our experimental input...” The tentative cutoff was suggested by IR Biotyper software? Explain better how the tentative cutoff value was defined. It would be interesting to add this information in the methodology.

Reviewer_1

“We showed that IR Biotyper was comparable to the STR method, which is still considered the method of choice“ – seems to be an inadequate and very strong statement since the results showed 47 to 74% agreement. In my opinion, a comparison with WGS would be justified here. Otherwise, the results are not very convincing.

Answer: We agree with the reviewer that the above statement may be too strong considering that the agreement of IR Biotyper with the STR method is not greater than 74%. Accordingly, we have edited the statement to mitigate its strength. See page 2 of the revised manuscript.

We also agree that using WGS to validate the comparison of IR Biotyper with the STR method in this study could have strengthened our findings. Therefore, we have added a sentence that emphasizes the limitation of our study. See page 9 of the revised manuscript.

Results: The authors should provide characteristic spectra for *C. parapsilosis* and discuss the spectral changes between fluconazole-susceptible and fluconazole-resistant strains.

Answer: As requested, we have added a new Figure (Fig. 4) showing the spectral profiles of representative fluconazole-resistant and fluconazole-susceptible *C. parapsilosis* isolates. As shown the profiles overlap and only subtle differences in absorbance are visible. See Fig. 4, and pages 6 and 18 of the revised manuscript.

Methods: How were the replicates treated? Were the spectra averaged, or do the results show individual results? Were there any differences among the replicates?

Answer: As requested, we have provided all information on how the replicates were treated or how an average spectrum was obtained. See page 11 of the revised manuscript.

Was the ability to form biofilms on solid artificial surfaces tested? It was shown that biofilm formation changes the FTIR spectra, and it is well-known that some *C. parapsilosis* strains have this ability.

Answer: While agreeing that biofilm formation may change FTIR spectra, isolates in our study were not tested while growing in biofilms. We have previously tested some *C. parapsilosis* isolates from another collection, but we did not observe any difference in FTIR spectra between high or low biofilm forming isolates. Thus, it is plausible that biofilm-related cell surface components do not influence the FTIR analysis results in our study.

It is not clear how the tentative value was determined. I believe it is crucial since it improved the results so significantly.

Answer: As requested we have added details on how the tentative cutoff value was determined. See page 11 of the revised manuscript.

Reviewer_2

The article “The Fourier-Transform Infrared Spectroscopy Based Method as a New Typing Tool for *Candida parapsilosis* Clinical Isolates” is relevant, considering the importance of developing techniques for typing microorganisms that are faster and more appropriate in the hospital context. Methods based on spectroscopy are particularly relevant, considering the advantages and numerous possibilities of these techniques. The authors showed that the IR Biotyper is capable of detecting differences in isolates considered clonal by the STR, increasing its discriminatory power in relation to the standard method.

The work is written concisely and clearly and is interesting and relevant. However, I have some aspects to point out that needs to be better explored in the writing, mainly in the Materials and Methods section.

Answer: We are grateful to the reviewer for appreciating this work and for giving us the opportunity to modify the manuscript based on her/his comments/suggestions.

1) Line 139: “To our knowledge, only one study to date has investigated IR Biotyper as a typing tool for *Candida* species (25)”. Please include the following article in the Discussion: Contreras DA, Morgan MA. Surveillance diagnostic algorithm using real-time PCR assay and strain typing method development to assist with the control of *C. auris* amid COVID-19 pandemic. *Front Cell Infect Microbiol.* 2022;12:887754. doi: 10.3389/fcimb.2022.887754. In this article, the authors used the IR Biotyper to genotype 28 hospital isolates of *C. auris*, using whole genome sequencing as a reference method. In addition, they were also able to use the IR Biotyper to successfully differentiate several yeast species, such as *C. albicans*, *C. glabrata*, *C. auris*, and *Cryptococcus neoformans*.

Answer: We apologize for this omission and now have cited (see page 5 of the revised manuscript) and discussed the findings (see page 7 of the revised manuscript) of the article.

2) Line 238: “Spectra were acquired (in the 1300–800 cm⁻¹ carbohydrate region) using the IR Biotyper system...”

I know that this is the default of the instrument and the “fingerprint” region. However, considering that the maximum Simpson's index of diversity value obtained was 0.839, was it considered that analyzing other regions of the spectrum could improve this index?

Answer: As suggested, spectra were analyzed in other IR spectral windows within the mid IR (4000–400 cm⁻¹), but results obtained confirmed the 1300–800 cm⁻¹ spectral region as the main contributor to the discriminatory potential of the IR Biotyper method. This was also in accordance with previous findings (see added ref. 34). See pages 11 and 12 of the revised manuscript.

3) Was any kind of pre-processing used in the spectra (normalization, derivatives...)? Could these pre-processes improve the discriminatory power of the IR Biotyper? I suggest better detailing the chemometric analyzes in the methodology.

Answer: As suggested, we have added details on the mathematical preprocessing steps for calculating the distances between spectra and sorting them in a hierarchical cluster analysis. See page 11 of the revised manuscript.

4) Line 116: “Using a tentative cutoff value (0.995), which was based on our experimental input...” Was the tentative cutoff suggested by IR Biotyper software? Explain better how the tentative cutoff value was defined. It would be interesting to add this information in the methodology.

Answer: As requested, we have added details on how the tentative cutoff value was defined. See page 11 of the revised manuscript.

July 21, 2023

Prof. Maurizio Sanguinetti
Fondazione Policlinico Universitario Agostino Gemelli IRCCS
Microbiology
L.go A. Gemelli 8
Rome, RM 168
Italy

Re: Spectrum02388-23R1 (The Fourier-Transform Infrared Spectroscopy Based Method as a New Typing Tool for *Candida parapsilosis* Clinical Isolates)

Dear Prof. Maurizio Sanguinetti:

Your manuscript has been accepted, and I am forwarding it to the ASM Journals Department for publication. You will be notified when your proofs are ready to be viewed.

Sincerely,

Renato Kovacs
Editor, Microbiology Spectrum
